# Nicastrin-Like, a Novel Transmembrane Protein from *Trypanosoma cruzi* Associated to the Flagellar Pocket

**DOI:** 10.3390/microorganisms9081750

**Published:** 2021-08-17

**Authors:** Guilherme Curty Lechuga, Paloma Napoleão-Pêgo, Larissa Rodrigues Gomes, Andressa da Matta Durans, David William Provance, Salvatore Giovanni De-Simone

**Affiliations:** 1FIOCRUZ, Center for Technological Development in Health (CDTS), National Institute of Science and Technology for Innovation on Neglected Diseases Populations (INCT-IDPN), Rio de Janeiro 21040-900, Brazil; guilherme.lechuga@cdts.fiocruz.br (G.C.L.); paloma.pego@cdts.fiocruz.br (P.N.-P.); larissa.gomes@cdts.fiocruz.br (L.R.G.); andressa.durans@cdts.fiocruz.br (A.d.M.D.); bill.provance@cdts.fiocruz.br (D.W.P.J.); 2FIOCRUZ, Interdisciplinary Medical Research Laboratory, Oswaldo Cruz Institute, Rio de Janeiro 21040-900, Brazil; 3Department of Cellular and Molecular Biology, Biology Institute, Federal Fluminense University, Niterói 24020-141, Brazil

**Keywords:** *Trypanosoma cruzi*, nicastrin, presenilin, γ-secretase, transmembrane motif, B-cell epitope, SPOT synthesis, type I transmembrane glycoprotein, vesicular trafficking, flagellar pocket

## Abstract

Nicastrin (NICT) is a transmembrane protein physically associated with the polytypical aspartyl protease presenilin that plays a vital role in the correct localization and stabilization of presenilin to the membrane-bound γ-secretase complex. This complex is involved in the regulation of a wide range of cellular events, including cell signaling and the regulation of endocytosed membrane proteins for their trafficking and protein processing. Methods: In *Trypanosoma cruzi*, the causal agent of the Chagas disease, a NICT-like protein (Tc/NICT) was identified with a short C-terminus orthologous to the human protein, a large ectodomain (ECD) with numerous glycosylation sites and a single-core transmembrane domain containing a putative TM-domain (457GSVGA461) important for the γ-secretase complex activity. Results: Using the Spot-synthesis strategy with Chagasic patient sera, five extracellular epitopes were identified and synthetic forms were used to generate rabbit anti-Tc/NICT polyclonal serum that recognized a ~72-kDa molecule in immunoblots of *T. cruzi* epimastigote extracts. Confocal microscopy suggests that Tc/NICT is localized in the flagellar pocket, which is consistent with data from our previous studies with a *T. cruzi* presenilin-like protein. Phylogenetically, Tc/NICT was localized within a subgroup with the *T. rangeli* protein that is clearly detached from the other Trypanosomatidae, such as *T. brucei.* These results, together with a comparative analysis of the selected peptide sequence regions between the *T. cruzi* and mammalian proteins, suggest a divergence from the human NICT that might be relevant to Chagas disease pathology. As a whole, our data show that a NICT-like protein is expressed in the infective and replicative stages of *T. cruzi* and may be considered further evidence for a **γ**-secretase complex in trypanosomatids.

## 1. Introduction

Nicastrin (NICT) is a single-pass transmembrane glycoprotein that is part of the γ-secretase complex (GSC). This complex is present in all eukaryotic cells and functions in the maturation and processing of all type 1 proteins destined for cell membranes. In addition to NICT, the GSC contains three other compounds: presenilin (PS), anterior pharynx-defective 1 (Aph1) protein and presenilin enhancer 2 (PEN-2). Human nicastrin (hu-NICT) is physically associated with PS, with an important role in the stabilization and correct localization of PS to the membrane-bound GSC [1].

Presenilin is a unique protease in that it cleaves within the lipid bilayer targeting many type 1, single transmembrane-spanning proteins that vary widely in their sequence and size [2,3]. NICT, as a class 1 protein, is itself a substrate of presenilin and appears to have the function to recruit substrates that have had their large extracellular domains first removed by an upstream protease in a process termed “ectodomain shedding” [4]. This process generates a new, short extracellular stub with a free N-terminus, which is required for proteolysis by the PS, and hu-NICT is suggested to be involved in peptide recognition [5]. The hu-NICT also serves as a docking site for γ-secretase substrates such as amyloid precursor protein (APP) and Notch, directly binding to them for their proper presentation to γ-secretase PS to ensure the correct cleavage process [4,6].

The glycosylation of NICT appears to be critical for the substrate preference and enzymatic activity of γ-secretase [7]. As a glycosylated protein, NICT can present differences in the glycosylated sites that could allow it to assume other roles in the biochemical pathways that regulate Aβ length and response to γ-secretase modulators [8]. Furthermore, hu-NICT was suggested to be involved in different various other human pathologies [9], which include hidradenitis suppurativa [10,11], depigmentation and skin inflammation [12], as well as pancreatic cancer prognosis [13].

While much is known about the GSC compounds in superior eukaryotic organisms, very little is known about their role in inferior eukaryotic cells. Recently, we showed the potential for the presence of a GSC in *Trypanosoma cruzi* based on our characterization of a TM aspartyl protease that shows hallmarks of human PS [14]. *T. cruzi* is the causative agent for American trypanosomiasis, more commonly referred to as Chagas disease, and infections are typically spread by triatomine vectors that affect millions of people throughout Latin America [15]. In the present study, we mapped the B-cell epitopes of the putative gene for *T. cruzi* NICT and developed antibodies against synthetic peptide sequences to identify the cellular localization of the protein. The immunological importance of one of these epitopes was examined by parasite adhesion inhibition study and as a possible immunological target. Other structural studies and database searches revealed that the Tc/NICT contains a large, heavily glycosylated extracellular ectodomain with five linear B-cell epitopes and expresses an important putative TM-domain for the γ-secretase complex activity. Phylogenetic analysis showed that the Tc/NICT is localized in an own subgroup with the *T. rangeli.* In the search for new drug targets in *T. cruzi*, the GSC is a promising candidate.

## 2. Materials and Methods

### 2.1. Reagents

Amino-PEG500-UC540 cellulose membranes were obtained from Intavis AG Bioanalytical Instruments (Cologne, Germany). Amino acids for peptide synthesis were purchased from Calbiochem-Merck (Darmstadt, Germany). BSA, acetic anhydride, *N*,*N*-dimethylformamide, Freund’s incomplete adjuvant, DAPI, TRITC and FITC labeled anti-rabbit IgG antibodies, TRITC-phalloidin, monodancylcadaverine, maleimide activated kit, Tween^®^ 20, acetonitrile, monodancylcadaverine, tissue protease inhibitor cocktail and trifluoracetic acid were obtained from Sigma–Merck (St. Louis, MO, USA). Rabbit and goat alkaline phosphatase-labeled anti-human-IgG (AP-anti-huIgG) and anti-rabbit IgG (AP-anti-rabIgG) were purchased from Abcam (Cambridge, MA, USA). Super Signal R West Pico chemiluminescent substrate was from Pierce Biotechnology (Rockford, IL, USA). Centrifugal Filter Units (cut-off 10 kDa) were from Millipore (Bedford, MA, USA) and Nitro-Block II from Applied Biosystems, (Waltham, MA, USA). Fetal bovine serum (FBS) was from Thermo Fisher Scientific Inc. (Waltham, MA, USA) The brain and heart infusion (BHI) mediums were from Difco and nitrocellulose membrane from BioRad, (Hercules, CA, USA).

### 2.2. Parasites and Cell Culture

The CL-Brener strain of *T. cruzi* was obtained from Dr. Maria A. de Souza (Oswaldo Cruz Institute, FIOCRUZ, Rio de Janeiro, Brazil) and the Trypanosomatidae collection (CT–IOC/05). Parasites were maintained in BHI medium supplemented with 10% FBS [16]. Experimental conditions included the absence of FBS for 24 h or the addition of 50 µg/mL pepstatin A, a gamma secretase inhibitor. Trypomastigote forms were obtained from *T. cruzi*-infected Vero cell cultures, maintained in RPMI-1640 medium supplemented with 10% fetal bovine serum (FBS), at 4th day post-infection (dpi) [17].

### 2.3. Synthesis of the Cellulose-Membrane-Bound Peptide Array

The entire sequence (728aa) of the non-characterized protein nicastrin, deposited in the UNIPROT (Q4DEM3) and a possible component of GSC in *T. cruzi*, was represented by a continuous series of 144, 14-residue peptides that overlapped preceding and following peptides by 9 residues. The peptides were synthesized in place on a cellulose membrane according to a SPOT synthesis protocol [18] using an Auto-Spot Robot ASP-222 (Intavis Bioanalytical Instruments AG, Köln, Germany). Coupling reactions were followed by acetylation with acetic anhydride (4%, *v*/*v*) in *N*,*N*-dimethylformamide to render peptides unreactive during the subsequent steps. After acetylation, Fmoc protective groups were removed by the addition of piperidine to render nascent peptides reactive. The remaining amino acids were added by this same process of coupling, blocking and deprotection until the desired peptide was generated. After the addition of the last amino acid in the peptide, the amino acid side chains were deprotected using a solution of dichloromethane–trifluoracetic acid–triisobutylsilane (1:1:0.05, *v*/*v*/*v*) and washed with methanol. Membranes containing the synthetic peptides were either probed immediately or stored at −20 °C until needed. Negative and positive controls were included in each assay.

### 2.4. Screening of SPOT Membranes

SPOT membranes were washed with TBS (50 mM Tris-buffer saline, pH 7.0) and then blocked with TBS-MT (Tris-buffer saline, 3% defatted milk, 0.1% Tween^®^ 20, pH 7.0) at room temperature under agitation or overnight at 4 °C. After extensive washing with TBS-T (Tris-buffer saline, 0.1% Tween^®^ 20, pH 7.0), membranes were incubated for 2 h with a pool of human patient serum (*n* = 7) diluted 1:250 in TBS-MT and then washed again with TBS-T. After that, membranes were incubated with rabbit alkaline phosphatase labeled anti-human IgG (diluted 1:5000), prepared in TBS-CT for 1 h, and then washed with TBS-T and CBS (50 mM citrate-buffer saline). The chemiluminescent substrate Nitro-Block II was added to complete the reaction.

### 2.5. Scanning and Measurement of Spot Signal Intensities

Chemiluminescent signals were detected on an Odyssey FC (LI-COR Bioscience, Lincoln, NE, USA) as described previously [19]. Briefly, a digital image file was generated at a resolution of 5 MP, and the signal intensities were quantified using the TotalLab TL100 (v. 2009, Nonlinear Dynamics, Newcastle-Upon-Tyne, UK) software. The signal intensity (SI) used as a background was a set of negative controls spotted in each membrane.

### 2.6. Peptide Synthesis, BSA Conjugation and Polyclonal Antibodies Production

Two *T. cruzi* peptides, (Tc/NCT-1 = SLQDIIRGLSIPDT and Tc/NCT-4 = KSLRIPHGDW GATW), were chosen to be synthesized by the F-moc strategy in a synthesizer machine (PSS-8, Shimadzu, Kyoto, Japan) with a C-terminal cysteine that was used to conjugate peptides to bovine serum albumin (BSA) using a maleimide activated kit according to manufacturer’s instructions. The reaction mixture was passed through a centricon-P10, and the peptide concentration in the filtrate (uncoupled peptide) was measured on a Qubit device (Thermo Fisher, Waltham, MA, USA). After the coupling reaction, two New Zealand rabbits were immunized by subcutaneous injection of peptide-BSA (150 µg) emulsified with an equal volume of saponine. Three other inoculations, without adjuvant, were each administered 7 days later, and the serum was collected five days after the last injection. Blood was collected under standard bioethics conditions from the marginal ear vein.

### 2.7. Enzyme-Linked Immunosorbent Assay (ELISA)

Anti-Tc/NICT serology was performed on ELISA microplates (Immulon 2HB; Corning, NY, USA) as previously described [20]. In all experiments, background values were subtracted from the measurement tests. 

### 2.8. Preparation of Cell Extract

Detergent soluble fractions were prepared from *T. cruzi* epimastigotes (1 × 10^8^) in the log phase that was collected by centrifugation (5000× *g* for 30 min at 4 °C) and washed three times in PBS (pH 7.2). Parasites in the pellet were suspended in extraction buffer (150 mM NaCl, 50 mM Tris 50 (pH 7.5) with 1% Triton X-100 and protease inhibitors) and subjected to six cycles of freeze-thawing. The detergent soluble fraction in the supernatant was collected after centrifugation (100,000× *g* for 1 h at 4 °C). An extract of hydrophobic membrane proteins was obtained using a previously described method with adaptations [21]. Briefly, pre-condensed TritonX-114 (TX114) was used to make a lysis buffer with 2% TX114 in PBS with the addition of a protease inhibitor cocktail (Sigma, St. Louis, MO, USA). Epimastigotes (1 × 10^8^) were collected, washed and lysed as described above. Before centrifugation, lysis buffer was added, and samples were kept on ice (4 °C) for 30 min with vortexing every 10 s. The suspension was centrifuged (10,000× *g* for 10 min at 4 °C) to pellet cell debris, and the supernatant was incubated at 10 min at 37 °C to generate a phase separation by centrifugation (22,000× *g* for 10 min at 37 °C). The aqueous phase with hydrophilic proteins was carefully removed, and the detergent-rich phase was processed for phase separation two more times. Protein was precipitated from the final phase by the addition of 4 volumes of cold acetone (−20 °C) followed by vortexing and a 1 h incubation at −20 °C. The precipitate was pelleted by centrifugation (15,000 rpm for 30 min at 25 °C), and the pellets were air-dried before being solubilized in SDS-PAGE by a 5 min incubation at 100 °C. Protein concentration was measured using the Folin–Lowry method.

### 2.9. SDS-PAGE and Western Blotting

Protein samples were separated by sodium dodecyl sulfate (SDS)-polyacrylamide gels electrophoresis (SDS-PAGE, 10%) under reducing conditions [22]. For Western blotting, samples (30–40 μg of protein) were separated by SDS-PAGE and electro-blotted (35 min, 45 V) in a Bio-Rad transblot system onto nitrocellulose membrane (0.2 μm, Bio-Rad, Hercules, CA, USA) for 90 min. The transfer buffer contained 48 mM Tris, 39 mM glycine, 0.0375% (*w*/*v*) SDS and 20% (*v*/*v*) methanol. The nitrocellulose was saturated with 5% (*w*/*v*) non-fat powdered milk in TBS-T, washed three times with TBS-T, and incubated overnight with 1:200 dilution rabbit anti-Tc/NICT-1 sera. The sheets were washed three times with TBS-T, and immunoreactive proteins were localized using HRP-anti-rabIgG (1:5000) for 1 h at 25 °C. Proteins were detected by chemiluminescence using the SuperSignal West Pico substrate kit (Thermo Fisher, Waltham, MA, USA). Revealed bands were quantified by densitometry using Image J (http://rsbweb.nih.gov, accessed on 21 september 2020).

### 2.10. Identification of Glycoconjugates

*T. cruzi* epimastigotes extracts (detergent and soluble fractions) were obtained as described above and resolved by SDS-PAGE (10% under reducing conditions). After transfer to a nitrocellulose membrane, was used Immun-Blot^®^ Kit for glycoprotein detection (Bio-Rad, Hercules, CA, USA) following the manufacture’s instructions.

### 2.11. Immunofluorescence Microscopy

Parasites in the two different life stages, epimastigotes and trypomastigotes, were collected by centrifugation (5000× *g* for 30 min at 4 °C), washed twice with PBS, fixed with paraformaldehyde (4%) in suspension for 5 min and washed 3× in PBS. The parasites were placed onto a glass slide and allowed to air-dry overnight. Next, cells were blocked with 1% casein in PBS (pH 7.4) for 15 min at 37 °C. Afterward, parasites were incubated with rabbit anti-Tc/NICT-1 sera (1:100) for 2 h at 37 °C. Slides were washed in PBS and then incubated with TRITC-labeled anti-rabIgG (1:400) for 1 h at 37 °C. Additionally, anti-β-tubulin (1:400) was used to probe parasites’ microtubules. Following this, parasites were washed in PBS, stained with DAPI and mounted in DABCO solution. An Axio Imager M2 fluorescence microscope (Carl Zeiss) was used to collect images.

### 2.12. T. cruzi–Host Cell Interaction

Vero cells (5 × 10^4^) were maintained in supplemented RPMI 1640 medium (10% FBS and 2% l-glutamine) at 37 °C in a 5% CO_2_ atmosphere and seeded in 24-well culture plates containing 13 mm^2^ glass coverslips. For the analysis of parasite-host cell interactions, culture-derived trypomastigote forms (10^6^) of *T. cruzi* (CL strain) were incubated at 4 °C for 1 h with anti-Tc/NICT-1 sera (1:50; 1:100 and 1:200) or pre-immune sera (1:50) diluted into RPMI 1640 medium with 0.5% BSA. Next, parasites were added to plated Vero cells at 37 °C. After 2 h, cultures were washed three times with PBS to remove unbound parasites and fixed in Bouin’s solution, followed by staining with Giemsa. The attached parasites were counted from 100 cells in random fields of view. Counts were compared to the untreated group (Control). 

### 2.13. Phylogeny

The *T. cruzi* NICT sequence (Q4DEM3) was prospected in Blast (Delta-Blast) with a nucleotide database restricted Kinetoplast taxon. The evolutionary history was inferred by using the Maximum Likelihood method based on the JTT matrix-based model [23]. The initial tree(s) for the heuristic search were obtained automatically by applying Neighbor-Join and BioNJ algorithms to a matrix of pairwise distances estimated using a JTT model and then selecting the topology with a superior log-likelihood value. Evolutionary analyses were conducted in MEGA7 [24,25]. 

### 2.14. Bioinformatics Tools

The data bank search for *T. cruzi* NICT sequence (Q4DEM3) was performed using the human homologies to sequences already identified and annotated proteins in other organisms in the database UniProt (http://www.uniprot.org/, accessed on 21 August 2020) and nucleotide database restricted Kinetoplast taxon. The alignment of the sequences was performed using the T-Coffee server (http://tcoffee.vital-it.ch/cgi-bin/Tcoffee/tcoffee_cgi/index.cgi, accessed on 21 August 2020).

The hydropathicity scale of the protein was obtained with the ProtScale tool (http://web.expasy.org/protscale/, accessed on 15 January 2021) and TMHMM Server v. 2.0 (http://www.cbs.dtu.dk/services/TMHMM/, accessed on 15 January 2021). The prediction of the secondary structure of the protein was performed by the PSIPRED servers (http://bioinf.cs.ucl.ac.uk//, accessed on 10 February 2021) and CDM (http://gor.bb.iastate.edu/cdm/, accessed on 10 February 2021) and the tertiary structure prediction and the transmembrane domain (TM) by the I-TASSER server (https://zhanglab.ccmb.med.umich.edu/I-TASSER/, accessed on 15 November 2020). Some physicochemical parameters related to the peptides used in this study (molecular weight, theoretical pI and estimated hydrophobicity) were calculated with the ProtParam tool (http://web.expasy.org/protparam/, accessed on 12 November 2020). The protein transmembrane orientation was obtained by inserting protein structure in the orientation of membrane protein database (OMP, (https://opm.phar.umich.edu/, accessed on 12 November 2020)) using the PPM server [26]. Protein and epitope structure visualizations were performed using PyMOL (Molecular Graphics System, Version 2.0 Schrödinger, LLC). Helical wheel analyses of the transmembrane domain were obtained using Netwheels (http://lbqp.unb.br/NetWheels/, accessed on 10 August 2020). Moreover, the identification of nicastrin conserved motifs was performed on the MEME Suite server [27], which used sequences of nicastrin from different species curated in the Uniprot database (www.uniprot.org, accessed on 15 January 2021).

The predictions of glycosylation sites were realized using the NetNGlyc 1.0 Server (http://www.cbs.dtu.dk/services/NetNGlyc/, accessed on 15 January 2021). The N-GlyDE web server is available at http://bioapp.iis.sinica.edu.tw/N-GlyDE/, (accessed on 15 November 2020).

### 2.15. Animal, Human Sera and Ethics Statement

The approval for the experimental use of animals was granted by the Ethics Committee for Experimentation in Animals of the Oswaldo Cruz Foundation (CEUA P-0279/06) prior to the start of the study. The animals were housed and maintained according to the institutional guidelines for animal studies, which conform to the specifications set forth in the US National Institutes of Health guidelines for the care and use of laboratory animals. All efforts were made to minimize suffering.

Sera from patients chronically infected with *T. cruzi* (*n* = 92) were obtained from the Parasitic Disease Laboratory, Oswaldo Cruz Institute/FIOCRUZ (CAAE: 52892216.8.0000.5248 and 1.896.362). The diagnosis as positive for infections by *T. cruzi* was confirmed before inclusion in the study both clinically through the clinical signs and symptoms as well as serologically through immunofluorescence and enzyme-linked immunoassay (ELISA). For this study, patient sera remained anonymous and were pooled for experiments.

### 2.16. Statistical Analysis

Statistical analysis was performed using GraphPad Prism version 5.0. The statistical difference using a *t*-test was considered if *p*-value ≤ 0.05.

## 3. Results

### 3.1. Identification of the Immunodominant IgG Epitopes in T. cruzi Nicastrin

Epitopes in the putative Tc/NICT protein (728 aa) were identified based on recognition of peptides in a synthesized library (144 peptides) by Chagasic patient sera (see Materials and Methods). Figure 1A shows the image of the chemiluminescent signal detected from each peptide in the library for their reactivity with human IgG antibodies in sera pooled from infected patients. Positive peptides are designated by a box. Figure 1B presents the measured intensity and peptide position. Intensities were normalized to 100% as defined by the positive control (data not shown). A list of the peptides synthesized and their positions on the membranes are presented in Appendix A.

The pattern of reactivity for the antibodies generated in infected patients demonstrated that a significate number of peptides were recognized (Figure 1A). An analysis of the sequences constituting the peptides synthesized in reactive regions defined five major epitopes recognized by the sera of patients (Table 1).

### 3.2. Antigenicity of the Synthetic Peptides and Specificity of the Antisera 

When comparing the characteristics of the epitopes, two peptides, epitopes Tc/NICT-1 and Tc/NICT-4 were determined to display the most intense antigenic determinants and chosen for generating polyclonal sera. For immunizing rabbits, the peptides were synthesized with one cysteine at the carboxyl-terminal (LQLDIIRCLSIPDTC and KSLRIPHGDWGATWC) to facilitate covalent conjugation to BSA. Two rabbits were independently immunized for each peptide, and serum was monitored weakly. After the fifth (final) immunizations, the titer against Tc/NICT-1 increased ~3.3 times (dilution 1:40/1:80) in relation to the Tc/NICT-4 (Appendix A). The immunogenicity of the synthesized peptides was also evaluated by ELISA using antigen Tc/NICT-1 and Tc/NICT-4 MAPs and a panel of human Chagasic sera. The results are shown in Appendix A. 

The specificity of antisera against Tc/NICT-4 was analyzed by Western blot. A molecular mass of 79.7 kDa is predicted for the *T. cruzi* NICT protein. On Western blots, antibodies against the Tc/NICT-1 peptide detected in total extract only a single protein with a molecular mass of ~72 kDa (Figure 2A). In an evaluation for the presence of NICT in an enriched membrane fraction of hydrophobic proteins, anti-Tc/NICT-1 detected a protein with a molecular mass of ~92 kDa and a strong band at ~72 kDa. A faint band with a higher molecular mass was observed in the aqueous fraction (Figure 2C). The differences in the size indicated a possible maturation and glycosylation of NICT. Therefore, to investigate this hypothesis, glycoproteins were identified after SDS-PAGE and carbohydrate staining (Figure 2D). As expected, the detergent fraction containing membrane-enriched proteins presented intense staining showing two broad bands at ~150 kDa and ~95 kDa and one thin stained band at ~72 kDa. Interestingly, the same ~72 kDa glycosylated band was also seen in the insoluble fraction, suggesting that NICT partitioned the two phases and is expressed under the glycosylated form.

### 3.3. Subcellular Localization of the Tc-NICT 

The cellular localization of the *T. cruzi* NICT-like protein was determined by fluorescence microscopy using the anti-Tc/NICT-1 sera (Figure 3). Fluorescent images of Tc/NICT-1 staining showed a punctate staining pattern in the plasma membrane distributed along the body of trypomastigotes that are suggestive of distinct subdomains (Figure 3A). Epimastigote forms displayed a stronger staining signal for the Tc/NICT-1-like protein in the anterior region near the kinetoplast and flagellum (Figure 3B, arrowheads). The proximity of the signal to the flagellar pocket was seen as a strong signal in a divided parasite during the initial stages of duplication of the kinetoplast (Figure 3B). Additionally, to observed if Tc/NICT was in membrane microdomains, the parasite´s microtubules and subpellicular microtubules were probed with anti-β-tubulin. Punctate fluorescence of Tc/NICT was located above microtubules showing the presence of the protein in the trypomastigotes undulating membrane (Appendix A).

### 3.4. Nicastrin Participates in Cell Binding

To evaluate the participation of Tc/NICT in the interaction of parasites with host-cells, parasites were incubated with anti-Tc/NICT-1 for 1 h prior to their addition to cultures of Vero cells. Cells were exposed to antibody incubated parasites for 2 h at 4 °C to prevent internalization. Microscopic visualization of stained cells showed numerous trypomastigotes bound to Vero cells when left untreated or with pre-immune serum treated (1:50) parasites (Figure 4A,B, respectively). A reduction in the number of parasites attachment was noticeable with the highest concentration of anti-Tc/NICT-1 tested (1:50; Figure 4C). A graph of the quantification of bound parasites showed an increase of 1.5-fold for pre-immune serum compared to the untreated group (Figure 4D). Blocking Tc/NICT using anti-Tc/NICT-1 at 1:50 reduced proximally 28% and 54% parasites binding to the cellular surface compared to untreated or pre-immune serum treated group, respectively. Lower concentrations of anti-Tc/NICT-1 did not significantly alter *T. cruzi* attachment to Vero cells (Figure 4D).

### 3.5. Cross-Immunity

To investigate the cross-immunity conferred by the Tc/NICT protein across the lineages of *T. cruzi*, six sequences deposited in the NCBI data bank were aligned to compare the epitopes. This analysis showed that, in totality, all of the epitopes identified by sera from patients were identical to the different strain sequences of *T. cruzi* The high conservation of the structure of the epitopes suggests that a strong cross-immunity would be induced by immunization.

### 3.6. Structure and Topology of T. cruzi NICT-Like

All nicastrin family members described to date are predicted to be single-pass membrane proteins with one TMD. Using the TMHMM Server v.2.0, a single putative model was suggested for the *T. cruzi* NICT-like protein. This predicted structural model displayed a high probability of a TM domain at amino acids 704–716 (Figure 5B). With a single TMD, three well-defined segments were expected consisting of a long extracellular N-terminal extension (aa 1–703), a helical transmembrane segment (aa 704-LIFLMLSITATIS-716) and a very short cytoplasmic C-terminal portion (aa 727–728). In agreement with similar proteins, the beginning of the N-terminal (aa 1–38) displayed a hydrophobic profile (Figure 4B) that was consistent with a signal peptide chain that would be cleaved during processing and transported to the cell membrane, which would result in a mature protein comprising of the amino acids 39–728. 

The model presented in Figure 5A was generated using the I-Tasser platform. The analysis with the Orientation Membrane Protein access showed that some residues (aa 48–aa 54 and 58) could attach to the membrane. The localization of the epitopes identified by the SPOT synthesis analysis is also shown. Of the five epitopes identified, all were found in coil/loop structures of the NICT segment facing the extracellular plasma membrane (Figure 5A). The hydropathy plot of the protein and TM prediction also suggested that all the epitopes were present on the surface of the protein that faced the extracellular space. A helical wheel analysis of the predicted TM domain showed an alpha helix that was mostly composed of uncharged and unipolar amino acids (Figure 5C). 

### 3.7. Conserved Motifs

Data available from studies of GSC in animals indicate that certain amino acid motifs are crucial for proteolytic activity, substrate recognition, and complex assembly [28]. To verify if these patterns are also conserved in *T. cruzi*, multiple sequence alignments and motif identifications were conducted using amino acid sequences of NICT component homologs from *Dictyostelium discoideum*, *Physcomitrium patens*, *Chlamydomonas reinhardtii*, *Trypanosoma congolense*, *Trypanosoma cruzi*, *Trypanosoma brucei*, *Trypanosoma rangeli*, *Caenorhabditis elegans*, *Mus musculus*, *Arabidopsis thaliana*, *Chlamydomonas reinhardtii*, *Drosophila melanogaster and Homo sapiens*. Potential homologs were identified by PSI-BLAST, and similarities among investigated species in reference to *H. sapiens* sequences are presented in Figure 6. In superior eukaryotic cells, the motif DYIGS (aa 336–340) was shown to be critical for NICT function [29,30]. In *T.cruzi*, the motif was substituted by GSVGS (aa 402–406), in *T. rangeli* by GGVGS (aa 389–394), in *T. brucei* (aa 384–389) and *T. congolense* (aa 385–390) by GSIGS. In *Arabidopsis thaliana* by GYLGS (aa 402–407), in *Chlamydomonas* by GYMGS (aa 305–400), and in *Dictyostelium* by GYVGS (aa 300–305). 

The other domains and properties analyzed compared to human nicastrin are shown in Table 2. The TM segment of the Tc/NICT is located closer to the C-terminus, does not have the GGXXP domain, has a glutamic acid in a position closer to human nicastrin and has 2 embedde residues attached to the membrane.

### 3.8. Glycosylation 

*N*-linked glycosylation is one of the predominant post-translational modifications involved in a number of biological functions. Several predictors were made available, and most of them evaluate their performance at every asparagine in protein sequences but are not confined to asparagine in the N-X-S/T sequence. Using this approach, 7 *N*-glycosylation sites 84 (NNS), 329 (NET), 425 (NICT), 436 (NFT), 526 (NTT), 580 (NRS) and 675 (NDS) were identified in the *T. cruzi* NICT-like protein (Appendix A).

### 3.9. Phylogenies

A phylogenetic tree of NICT was retrieved on the Uniprot server (http://www.uniprot.org, accessed on 12 October 2020) search criteria name: “nicastrin” and length: 300 TO 800). Information of the sequence Q4DEM3 in Uniprot server shows “Uncharacterized protein” in protein name, and protein status is predicted with domains showing to be nicastrin. Prospection in Blast with Q4DEM3 sequence restricted in Kinetoplastidae taxon returned only the *Trypanosome* species. The tree with the highest log likelihood is shown in Figure 7. Mammalian NICTs were very closely related, and the comparison of human NICT with Trypanosome NICT showed a large divergence (data not shown). Cross identity, percentages varying 18.96% to 21.59%, as calculated with Muscle, are below the limit of homology. Searches on InterproScan [31] and HMMER [32] confirmed that the Q4DEM3 sequence is a NICT that showed a region between position 352 to 486 to be a NICT domain. Nicastrin seems to be a very divergent protein, maybe in this case assuming other roles or biochemical pathways.

## 4. Discussion

To characterize the putative *T. cruzi* NICT protein, a microarray of peptides was initially synthesized using the Spot-synthesis technique to identify linear B-cell epitopes recognized by Chagasic patient antibodies. Five epitopes were readily defined (Table 1), confirming the likelihood that the predicted *T. cruzi* NICT-like protein is expressed. Based on the molecular modeling of the sequences by three different computational algorithms, all the five epitopes are expected to be located in the extracellular domain. The best model for the *T. cruzi* NICT-like membrane insertion indicated the presence of only one TM domain (aa 704–726), similar to other orthologous proteins. This protein possesses 18 cysteine amino acids that may indicate a complex extracellular structure dependent on SH bridges. Interestingly, none of the 18 Cys amino acids were located within the identified epitope structures.

The model also suggested that each epitope consisted of random coils and turns with no definable secondary structures. Based on a physical-chemical analysis of stability, net charge at pH 7.0 and hydrophobicity, which contribute positively to antibody production, two epitopes (Tc/NICT-1 and Tc/NICT-4) were chosen to be synthesized as four-component MAP for antiserum production. In addition, each peptide showed high reactivity against Chagasic patient sera by peptide-ELISAs (Appendix A). The choice of peptides as antigens accelerated the antibody generation process by avoiding the need to purify the endogenous transmembrane protein, which can be laborious and low-yielding.

To test our hypothesis that the Tc/NICT is a membrane-associated protein, immunoblotting was performed with monospecific polyclonal sera anti-Tc/NICT-1. These polyclonal sera were chosen over the anti-Tc/NICT-4 based on its consistent reactivity at higher dilutions in an ELISA assay (Appendix A). The Western blot of the total extract using the rabbit sera anti-Tc/NICT-1 revealed a single band of approximately 72 kDa, a value compatible with the predicted molecular weight of the entire protein without glycosylation (75.9 kDa). However, as the mature protein most likely lost its signal peptide (aa 1–38), it suggests that the protein has approximately 3.9–10 kDa in carbohydrates. To enrich for the NICT protein, an extract of *T. cruzi* was subjected to phase separation using Triton-X114. Western blot analysis of hydrophobic proteins revealed a band at ~92 kDa suggesting the final mature form of Tc/NICT is highly glycosylated. This is consistent with the prediction for the presence of 7 *N*-glycosylation sites Tc-NICT protein has sites and explains the observations by immunoblot and molecular mass determination (Figure 2A,C). In addition, reinforcing in silico analysis, we identified in a membrane-enriched fraction a highly glycosylated broad band at ~95 kDa that matches the approximately predicted size of the band identified in Western blot (~92 kDa). Moreover, a thin band of glycoprotein was identified at ~72 kDa in detergent and soluble fractions, which could indicate that TcNICT has different glycosylated states as occur with human NICT [34]. Although the importance of these glycosylation sites is not fully understood. It was observed that hu/NICT N-linked oligosaccharides mediate specific interactions with the secretory pathway lectins calnexin and ERGIC-53 [35]. Further, the glycosylation status appears to drive the catalytic activity and substrate preference of the gamma-secretase complex [7].

Here, the subcellular localization of *T. cruzi* NICT-like protein was determined using the anti-Tc/NICT-1 polyclonal serum and fluorescence microscopy on multiple forms of the parasite in the absence of cell permeabilization. An apparent surface concentration of signal for the protease was observed in the anterior region of epimastigotes near the kinetoplast. This corresponds to the flagellar pocket, an area important for parasite nutrition and other cellular processes marked by a small invagination of the plasma membrane where the flagellum exits the cytoplasm. The punctate intracellular signals, together with its fractionation to membranes, suggest that the NICT-like protein could also be localized to the endoplasmic reticulum, Golgi complex and mitochondria similar to the *T. cruzi* presenilin localization [14]. In other cellular systems, the integration of these organelles with the PS/γ-secretase complex is responsible for aspects of the secretory pathways [36,37]. In *T. cruzi* and other trypanosomatids, the secretory pathway involves the endoplasmic reticulum and Golgi complex to the flagellar pocket, which is the main site of exocytosis and endocytosis. Together, they are part of a multi-organelle complex that has was implicated in cell polarity and division [38]. Interestingly, the strong fluorescence signal in epimastigote under division may represent a conserved function of the gamma-secretase complex, involved in the proteolytic activity and in the case of NICT substrate recognition [4,39]. 

Its localization to the flagellar pocket and parasite’s membrane may also indicate that it is involved in a compartment of the cell where an intense endocytic/secretory activity of proteins occurs. In eukaryotic cells, the activity of PS and γ-secretase complex was associated with intracellular trafficking and recycling of endosomal soluble proteins and membrane-associated receptors, such as transferrin receptors, through the endocytic recycling compartment [40]. In *T. cruzi*, transferrin receptors are expressed and localized in the flagellar pocket of epimastigotes and amastigotes. After transferrin binding, endosomes deliver it to reservosomes [41]. As epimastigotes are replicative forms, it would be expected that they would have a larger demand for molecules to sustain proliferation. Recently, we showed that the replicative forms of *T. cruzi* have the highest levels of PS-like signals, and these results were associated with a potential role for a γ-secretase complex in functions related to proliferation and protein maturation [14]. 

Conversely, NICT may be involved in regulating intracellular protein trafficking of the nascent presenilin complex during its assembly [1,42,43] and in binding to the N-terminus of substrates [1,38,42]. However, NICT is not essential for γ-secretase activity [41]. Nevertheless, evidence suggests a role of NICT as a substrate receptor [44], so we hypothesized that blocking its interaction by antibody binding could impair parasite host-cell interaction. The binding of *T. cruzi* to host cells is a complex mechanism that could involve many transmembrane receptors, extracellular matrix and proteases release [45]. Treatment of trypomastigotes with pepstatin an inhibitor of GSC reduced parasite adhesion to fibroblast by approximately 50 % [46]. However, the blocking of nicastrin, using anti-Tc/NICT-1, was not sufficient to produce a relevant inhibition, although when compared to pre-immune serum, we observed a 54% reduction at the highest antibody titer. 

The phylogenetic analysis showed that the *T. cruzi* and vertebrate homologs were separated into divergent clades (Figure 7), and the comparative structural studies carried out indicated that there are some significate structural differences between the human and the *T. cruzi* orthologous protein, a fact that could entail functional differences. The hu/NICT motif DYIGS is critical for substrate recognition and processing in γ-secretase. This motif in *T. cruzi* and *T. brucei* shared the sequence GSXGS (*T. cruzi*, 457-GSVGS-461). The change in amino acids between I, L, M, and V should not affect the hydrophobic character of the domain. However, the 457D to G substitution could have a profound influence on the polar characteristics of this domain in Tc/NICT. Nevertheless, experiments performed in *Dictyostelium* [47], which has a similar domain sequence, indicated that γ-secretase was still proteolytically active against mammalian substrates, and therefore one can conclude that this substitution did not necessarily abolish complex activity. 

Another important structural residue implicated in the γ-secretase activity of human cells [48] is Glu-333. This amino acid in the Tc/NICT protein is located at the position Glu-330 and thus accessible to the molecular surface, similar to the human form. This family of proteins is also characterized by a TM domain in the C-terminal segment. In *T. cruzi*, the TM (aa 704–716) is localized more distal than the TM helix of the hu-NICT (aa 667–690), but both contain a hydrophilic patch that is important for TM–TM interactions. The short C-terminus appeared not to be structured, which was also see in the Tc/NICT similarly found for others NICT proteins [49]. In the hu/NICT, a region formed by residues V697–A702 was identified that interacts with membranes and appears to play a role in the γ-secretase complex formation [50,51]. In contrast, Tc/NICT appeared to have an embedded membrane association found in the N-terminus aa 48–54) and aa 58 (Figure 5). 

A major difference observed between hu/NICT and Tc/NICT was related to the number of predicted glycosylation sites present in the ectodomain (EDC). While the human presents 16 sites (~36 kDa total mass) and adopts a thermostable structure [52], the Tc/NICT possessed only eight sites. It is known that protein glycosylation can influence a variety of biological processes such as protein folding, signaling, trafficking, cell–cell interactions and immune response [53]. In the case of hu/NICT, previous observations suggest that the ECD might be involved in substrate selection and acquisition [39]. Therefore, at this moment, functional consequences of these structural differences cannot be defined for the three Tc/NICT proteins in the GSC.

This is the first description of a NICT orthologue protein in *T. cruzi* that includes its cellular localization and modeling. While the function of the *T. cruzi* NICT-like enzyme is not yet defined, our results show that it is expressed, and its subcellular localization is suggestive for a function in secretion and/or endocytic trafficking. The identification of NICT as a bona fide protein further supports the possibility of the formation of GSC in parasites. Antibody blocking studies suggest that it could play a role in parasite–host cell interactions. An intensive investigation of the spatial and temporal expression patterns of the *T. cruzi* NICT-like gene combined with the use of new knockout techniques should help reveal their relevant function in the host–parasite relationship.

## 5. Conclusions

We demonstrated that *T. cruzi* expresses a nicastrin protein with structural properties similar to its orthologous human protein. The protein is predicted to have a single transmembrane domain with a long extracellular N-terminus that was corroborated by localization studies. Five IgG-binding epitopes were restricted to unfolded regions in the extracellular segment according to the structural model. Antibodies against Tc/NICT do not interfere significantly with the growth and viability of the parasite but can interfere with parasite binding mammalian cells. Structural analysis revealed that the motif TM of the nicastrin is conserved in trypanosomatids, which suggests there could be some function conservation regardless of the structural differences identified. This is the first study to describe the cellular localization, the partial structural characterization and the specific amino acid sequences involved in the immune recognition of Tc/NICT. Overall, our data should advance the understanding of the role of the y-secretase complex in the *Trypanosoma cruzi.*

## Figures and Tables

**Figure 1 microorganisms-09-01750-f001:**
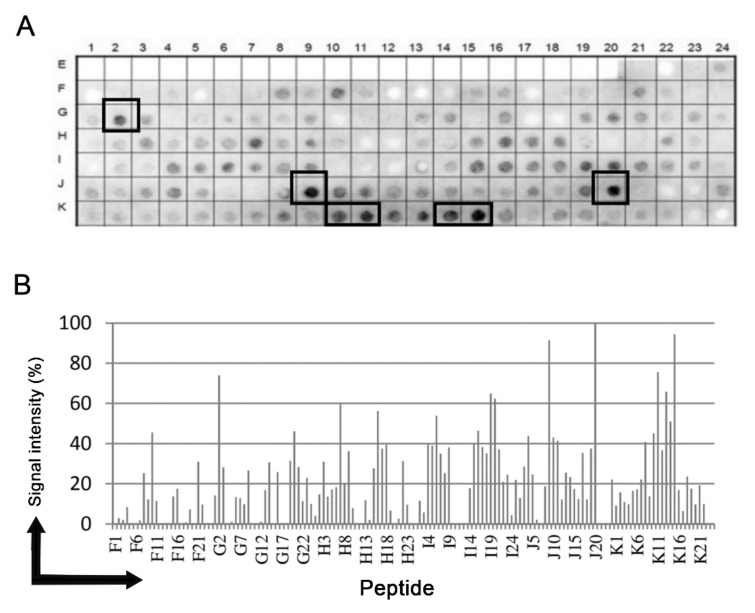
Binding of human IgG to a cellulose-bound peptide library representing the putative NICT *T. cruzi* protein. An overlapping array of 144 peptides, shifted 5 residues from each other, were probed with a 1:250 dilution of a pool of human sera and human IgG was detected by alkaline phosphatase labeled rabbit anti-human IgG and chemiluminescence. (**A**) Image of the membrane showing the reactivity at each spot and the positions used to make the measurements presented in panel (**A**). (**B**) Relative signal intensity of bound human IgG to each position from the membrane. 100% was defined by the positive control and 0% by the negative control. A list of the individual peptides spanning the *T. cruzi* uncharacterized protein (Q4DEM3) and constitute the library with their positions on the membrane is shown in the Appendix A.

**Figure 2 microorganisms-09-01750-f002:**
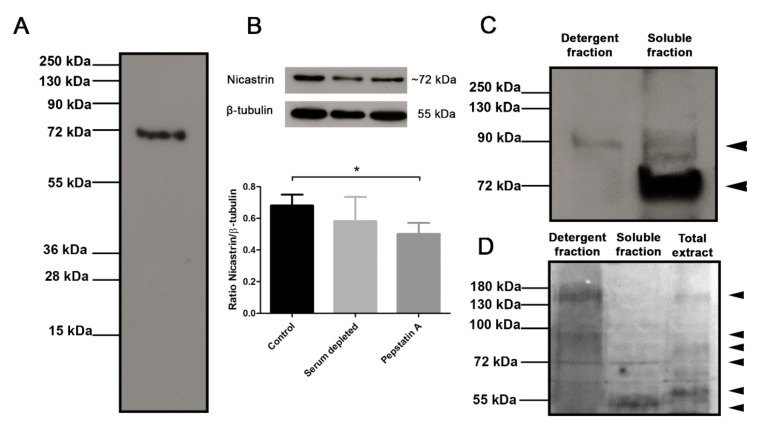
Immunoblot analysis of anti-Tc/NICT-1 rabbit polyclonal serum for detection of the *T. cruzi* nicastrin-like protein. (**A**) The whole extract (20 µg) probed with anti-Tc/NICT-1 serum identified a band with ~72 kDa. (**B**) Whole-cell extracts (20 µg) from epimastigotes cultured 24 h under serum deprivation (BHI medium without FBS) and after treatment with gamma-secretase inhibitor pepstatin A (50 µg/mL) probed with polyclonal anti-Tc/NICT-1 serum or anti-β-tubulin antibodies (internal control). The corresponding densitometry presents the ratio of *T. cruzi* NICT-like protein and β-tubulin. Data represent the mean and standard deviation from at least three independent experiments. (**C**) Membrane enriched fraction and soluble proteins probed with anti-Tc/NICT-1. (**D**) Detection of glycoproteins in *T. cruzi* epimastigote extracts after SDS-PAGE separation of detergent fraction (20 µg), the soluble fraction (20 µg), and total extract (20 µg). Stained bands were developed using a commercial kit (Immun-Blot^®^ Kit for glycoprotein detection (Bio-Rad)) and demonstrate glycosylated proteins in bands. Six different markedly bands (arrowheads) were observed in the detergent fraction (~150, ~95, and ~72 kDa), the soluble fraction (~72 and ~50 kDa), and epimastigote whole extract (~150, ~80, and ~60 kDa). * Significant difference using *t*-test (*p* < 0.05) (for original blotting see Appendix A).

**Figure 3 microorganisms-09-01750-f003:**
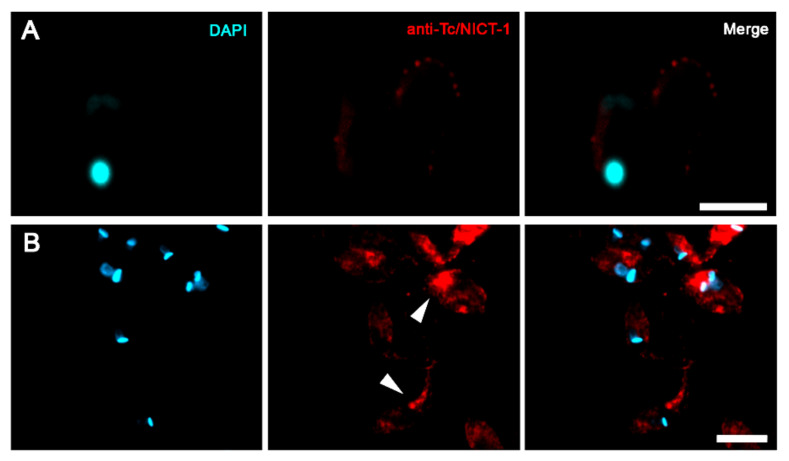
Immunofluorescent localization of *T. cruzi* nicastrin-like protein in trypomastigotes and epimastigotes. Cultures of trypomastigotes (**A**) and epimastigotes (**B**) were labeled with DAPI (cyan) and anti-Tc/NICT-1 (red). The merged images show the punctate membrane localization in trypomastigotes and a concentration of signal near the flagellar pocket (white arrowed) in individual epimastigotes or dividing cells. Scale bar = 10 µm.

**Figure 4 microorganisms-09-01750-f004:**
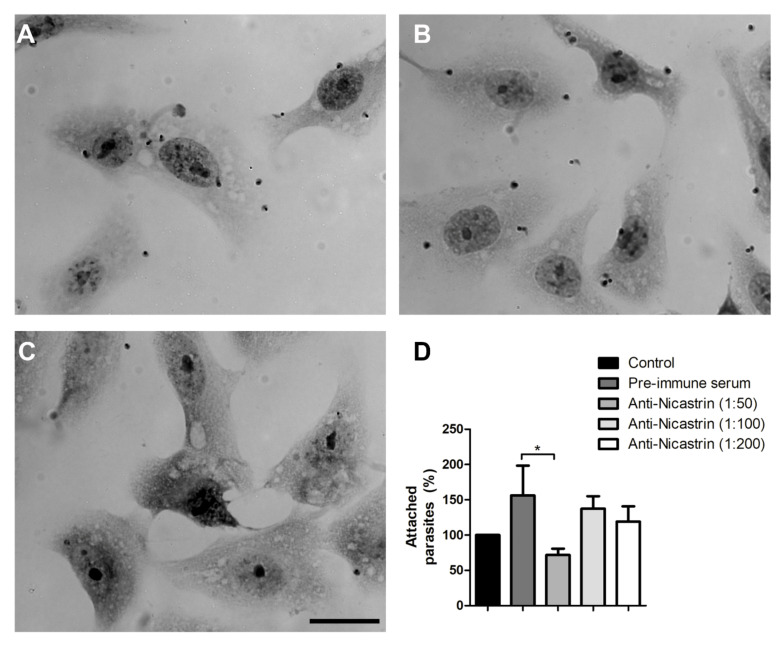
Analysis of host cell-trypomastigote interactions after nicastrin blocking using anti-Tc/NICT-1 sera. Light microscopy images of *T. cruzi* attached to Vero cells 2 h after parasite addition. Trypomastigotes were pre-incubated for 1 h with medium (**A**), pre-immune rabbit sera (**B**) or anti-Tc/NICT-1 sera at a dilution of 1:50 (**C**). The number of trypomastigotes attached to the cellular surface was quantified, and the percentage of bound parasites was calculated using the untreated group as 100% (Control). (**D**) The number of parasites was determined by random quantification of at least 100 cells from each replicate. * Significant difference using *t*-test (*p* < 0.05). Scale bar = 20 µm.

**Figure 5 microorganisms-09-01750-f005:**
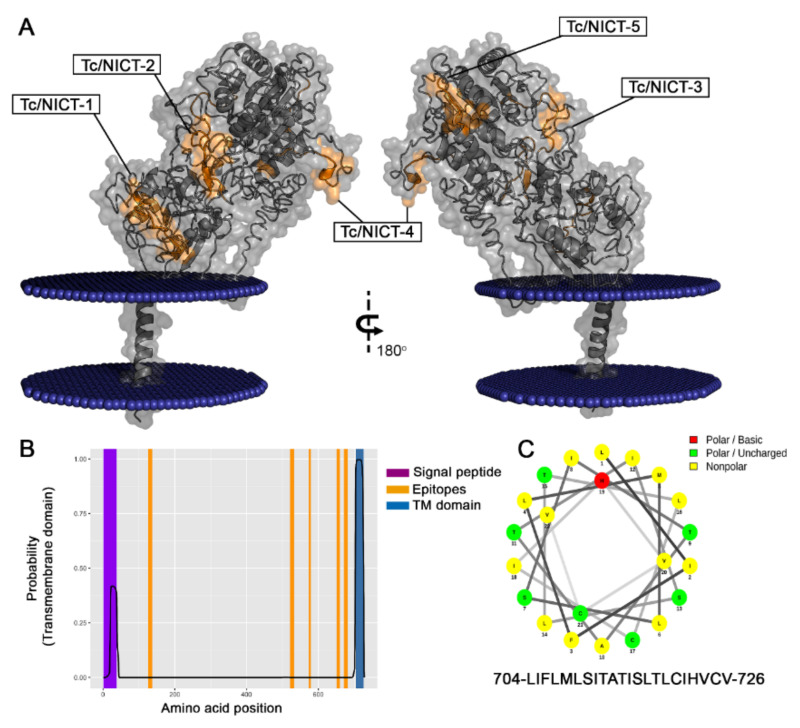
Model of Tc-NICT with amino acid and transmembrane domain analysis. (**A**) Putative membrane-spanning model of *Trypanosoma cruzi* NICT with the TM domain (aa 704–aa 726), a large extracellular domain, and a site of membrane interaction with embedded residues (aa 48–54, aa 58) determined using the I-Tasser approach Orientation Membrane Protein access (https://opm.phar.umich.edu/, accessed on 1 October 2020). The five linear B epitopes (Orange) identified by Spot-synthesis are shown. (**B**) Hydropathy plot showing the TM probability (blue) of NICT amino acid sequences along with the signal sequence (purple) and epitopes (orange). (**C**) Helical wheel analysis of TM domain showing the amino acid composition of an alpha helix.

**Figure 6 microorganisms-09-01750-f006:**
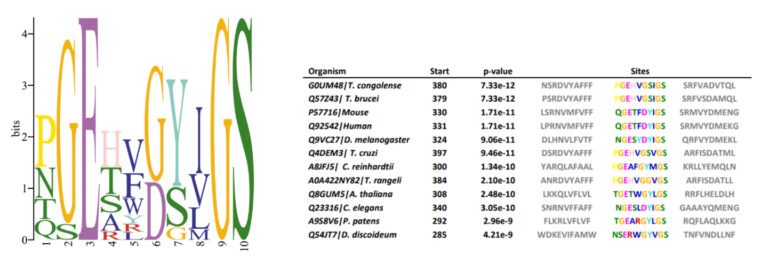
Sequence logo of nicastrin conserved motif. Multiple sequence alignment of different organisms (*T. congolense*, *T. cruzi*, *T. brucei*, *T. rangeli*, *C. elegans*, *D. discoideum*, *P. patens*, *Chlamydomonas reinhardtii*, *M. musculus*, *A. thaliana*, *C. reinhardtii*, *D. melanogaster* and *H. sapiens*) show amino acid variations in nicastrin motif DYIGS.

**Figure 7 microorganisms-09-01750-f007:**
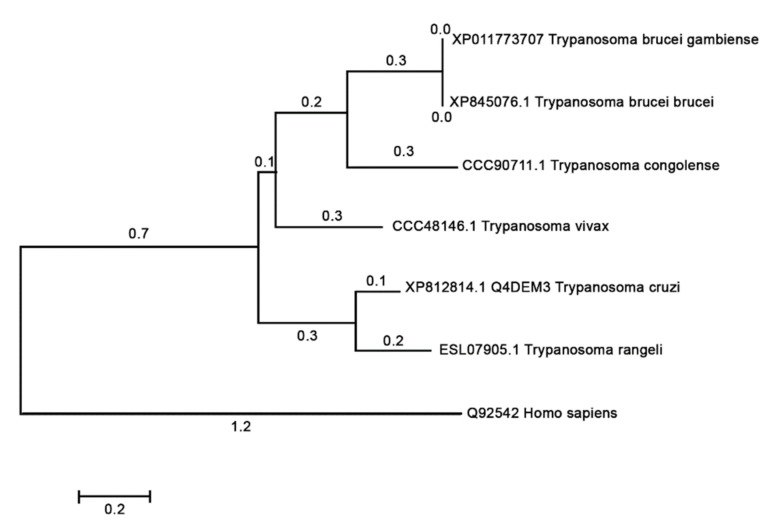
Phylogenetic relationship of *T. cruzi* nicastrin. The evolutionary history was inferred by using the Maximum Likelihood method based on the JTT matrix-based model [33]. The tree with the highest log likelihood is shown. Initial tree(s) for the heuristic search were obtained automatically by applying Neighbor-Join and BioNJ algorithms to a matrix of pairwise distances estimated using a JTT model, and then selecting the topology with superior log likelihood value. Evolutionary analyses were conducted in MEGA7 [25].

**Table 1 microorganisms-09-01750-t001:** Epitopes mapped in the *Trypanosoma cruzi* NICT-like protein (Q4DEM3) using a pool of sera from patients with chronic Chagas disease.

Epitope	Sequence	aa
Tc/NICT-1	SLQDIIRGLSIPDT	126–139
Tc/NICT-2	TLTRYNTTFANPDV	521–534
Tc/NICT-3	VTSVNR	576–581
Tx/NICT-4	KSLRIPHGDWGATW	651–664
Tc/NICT-5	MRLHNDSRYELHVM	671–684

**Table 2 microorganisms-09-01750-t002:** Comparison of domains and properties within *T. cruzi* and human cells NICT protein.

Domain/Properties	*H. sapiens* (Q92542)	*T. cruzi* (Q4DEM3)
Signal peptide	1–38	1–38
Chain	38–728	38–703
Embedded residues	-	48–54, 58
TM domain	671–690	704–726
Glu, as important residue for gamma-secretase activity	333	330
Motif GGXXP	300–304	ND
Motif DYIGS	DYIGS (336–340)	GSVGS (402–406)
Try previous the conserved domains aa 352–486	Yes	Yes

ND: not determined.

## Data Availability

The data presented in this study are available in Appendix A and on request from the corresponding author.

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
