# Peer review of "Nicastrin-Like, a Novel Transmembrane Protein from Trypanosoma cruzi Associated to the Flagellar Pocket"

_microorganisms, 2021, doi:10.3390/microorganisms9081750_

Round 1
Reviewer 1 Report
The manuscript is very interesting. The experiments and the results are well described, and the conclusions presented are supported by these results.
Minor comments:
1. All scientific names of pathogens should be written in italics
2. Remove underlines in the text.
3. There are minor english errors throughout the manuscript
Author Response
The manuscript is very interesting. The experiments and the results are well described, and the conclusions presented are supported by these results.
Minor comments:
- All scientific names of pathogens should be written in italics
- Remove underlines in the text.
- There are minor English errors throughout the manuscript
R: We appreciate the work and effort of the reviewers who brought essential corrections to the text. All revisions were accepted and modified in the text. In addition, the manuscript has been revised by a native English speaker.

Reviewer 2 Report
The works by Lechuga and colleagues is interesting, in their working strategy they took a reverse approach to the one used for most molecular biology studies. Authors most clarify many issues, such the molecularr weight differences of the protein when detected in Western blots assays, and perhaps a visual model could help to understand this point.
Second: the IFA results are not quite clear the fluorescent photos are of poor quality, suggesting a not very specific localization, the protein is all over the cell.
Third: the authors suggest that the protein undergoes glycosylation but providing no evidence. This is an importan point that needs to be addressed in order to make sense of all discrepancies in MW and potential fuction of the protein. In the annexed PDF of the work I marked in yellow important points, and some where there are typos and doubts

Author Response
Referee 2
The work by Lechuga and colleagues is interesting, in their working strategy, they took a reverse approach to the one used for most molecular biology studies.
We appreciate the reviewer comments who brought essential corrections to the text. All revisions were carefully checked, accepted, and modified in the text.
1)Authors must clarify many issues, such as the molecular weight differences of the protein when detected in Western blot assays, and perhaps a visual model could help understand this point.
R: The different molecular weights identified match the estimated molecular size of TcNICT (~79 kDa). The western blot band of ~72 kDa represents the protein weight without the signal peptide. Also, it is well described in mammalian cells that nicastrin undergo posttranslational N-glycosylation during trafficking from the endoplasmic reticulum (ER) to the Golgi apparatus. The presence of glycosylation in nicastrin significantly alters the electrophoretic mobility in SDS-PAGE. Please, see the article below. So, we hypothesized that higher molecular weight bands (~92 kDa) identified in western blot is the mature nicastrin, as described in humans.
2)Second: the IFA results are not entirely clear; the fluorescent photos are of poor quality, suggesting a not very specific localization; the protein is all over the cell.
R: Nicastrin is synthesized in the endoplasmatic reticulum (ER), glycosylated in Golgi apparatus, and then in vesicles transported to the membrane. So, it is expected to find immunofluorescence in parasites membrane, vesicles, and ER. Our previous work characterizing TcPresenilin, another component of the Gamma Secretase Complex, showed similar subcellular immunostaining. We applied different probes to stain ER (DiOC6), membrane (ConA), and acidic vesicles (PepstatinA-BODIPY) and found specially colocalization of TcPresenilin with ER and membrane. Equally, our previous work, with presenilin, showed a strong fluorescent signal in the flagellar pocket, a specialized membrane microdomain involved in the endocytic pathway and many biochemical processes. We addressed a new figure in the supplemental material (Fig. S4), showing that in trypomastigotes, the infective form, the immunostaining of TcNICT, is above the subpellicular microtubules, demonstrating the presence of the protein in membrane microdomains and undulating membrane.
In the discussion, we briefly comment on these findings:
Line 510: An apparent surface concentration of signal for the protease was observed in the anterior region of epimastigotes near the kinetoplast. This localization corresponds to the flagellar pocket, an area essential for parasite nutrition and other cellular processes marked by a small invagination of the plasma membrane where the flagellum exits the cytoplasm. In addition, the punctate intracellular signals, together with its fractionation to membranes, suggesting that the NICT-like protein could also be localized to the endoplasmic reticulum, Golgi complex, and mitochondria similar to the T. cruzi presenilin localization [14].
3) Third: the authors suggest that the protein undergoes glycosylation but providing no evidence. This is an important point that needs to be addressed to make sense of all discrepancies in MW and the potential function of the protein. In the annexed PDF of the work, I marked in important yellow points, and somewhere there are typos and doubts
R: Initially, we performed a silico analysis to find possible glycosylation sites. We agree with the referee that a silico study is not strong evidence of the biological event. So, we decided to perform another assay for the identification of glycoprotein in parasite extracts. Using membrane enriched, soluble, and total extract, we identified several bands that contain glycoproteins, including the ~72 kDa and broadband of ~95 kDa in the detergent fraction that matched the recognized band in western blot. A new figure (Fig. 2D) was added.
We addressed in the manuscript a discussion of these findings.
Line 500: In addition, reinforcing in silico analysis, we identified in a membrane-enriched fraction highly glycosylated broadband at ~95 kDa that match the approximately predicted size of the band identified in western blot (~92 kDa). Also, a thin band of glycoprotein was identified at ~72 kDa in detergent and soluble fractions, indicating that TcNICT has different glycosylated states as occur with human NICT [34].
Other corrections addressed to the manuscript:
Line 61: While much is known about the GSC compounds in complex eukaryotic organisms, very little is known about their role in less complicated eukaryotic microorganisms.
Line 197: derived trypomastigote forms (106 ) of T. cruzi (CL strain) were incubated at 4 °C for 1h
Line 216: Removed the t. “…nucleotide database restricted Kinetoplast taxon…."
Section 3.2. Antigenicity of the Synthetic Peptides and Specificity of the Antisera
Molecular masses were checked. “….The specificity of antisera against Tc/NICT-4 was analyzed by western blot. A molecular mass of 79.7 kDa is predicted for the T. cruzi NICT protein. On western blots, antibodies against the Tc/NICT-1 peptide detected in total extract only a single protein with a molecular weight of approximately 72 kDa (Fig. 2A). In an evaluation for the presence of NICT in an enriched membrane fraction of hydrophobic proteins, anti-Tc/NICT-1 detected a protein with a molecular weight of approximately 92 kDa and a strong band with at ap-proximately 72 kDa. In addition, a faint band with a higher molecular mass was observed in the aqueous fraction (Fig. 2C). The differences in the size indicated a possible maturation and glycosylation of NICT. Thus to test this hypothesis, glycoproteins were identified after protein electrophoresis observed in Supplementary Fig 3. As expected, detergent fractions containing membrane-enriched proteins presented intense staining showing two strongly stained broad bands at ~150 kDa and ~95 kDa and one thin stained band at ~72 kDa. Interestingly, the same ~72 kDa glycosylated band was also identified insoluble fraction, suggesting that NICT could be presented in the glycosylated form.”
Section: Discussion
“…The western blot of the total extract using the rabbit sera anti-Tc/NICT-1 revealed a single band of approximately 72 kDa, a value compatible with the predicted molecular weight of the complete protein without glycosylation (75.9 kDa). However, as the mature protein most likely lost its signal peptide (aa 1-38), it suggests that it has approximately 3.9-10 kDa in carbohydrates. The enrichment of the NICT protein was obtained using Triton-X114. Western blot analysis of hydrophobic proteins revealed a band at 92 kDa suggesting the final mature form of Tc/NICT is highly glycosylated. The prediction for 7 N-glycosylation sites Tc-NICT protein has sites and explains immunoblot and molecular mass determination (Figure 2 A and C). In addition, reinforcing in silico analysis, we identified in a membrane-enriched fraction highly glycosylated broadband at ~95 kDa that matches the approximately predicted size of the band identified in western blot (~92 kDa). Also, a thin band of glycoprotein was identified at ~72 kDa in detergent and soluble fractions, indicating that TcNICT has different glycosylated states as occur with human NICT [34]. Although the importance of these glycosylation sites is not fully understood. It has been observed that hu/NICT N-linked oligosaccharides mediate specific interactions with the secretory pathway lectins calnexin and ERGIC-53 [35]. Further, the glycosylation status appears to drive the catalytic activity and substrate preference of the gamma-secretase complex [7].

Round 2
Reviewer 2 Report
I believe the Authors have responded to my inquires, therefore I accept the MS, nevertheless, I will suggest improving de quality of the immunofluorescence, these are rather blurred